# The clinical effectiveness of one-dose vaccination with an HPV vaccine: A meta-analysis of 902,368 vaccinated women

Didik Setiawan[1,2]*, Nunuk Aries Nurulita[1], Sudewi Mukaromah Khoirunnisa[3,4], Maarten J. Postma[3,5,6]

1 Faculty of Pharmacy, University of Muhammadiyah Purwokerto, Purwokerto, Indonesia, 2 Center for Health Economic Studies, Universitas Muhammadiyah Purwokerto, Purwokerto, Indonesia, 3 Department of Health Sciences, University of Groningen, University Medical Center Groningen, Groningen, The Netherlands, 4 Department of Pharmacy, Institute Teknologi Sumatera, Lampung Selatan, Indonesia, 5 Unit of Pharmaco-Therapy, Epidemiology & Economics (PTE2), Department of Pharmacy, University of Groningen, Groningen, The Netherlands, 6 Department of Economics, Econometrics & Finance, Faculty of Economics & Business, University of Groningen, Groningen, The Netherlands

* d.didiksetiawan@gmail.com

**Data Availability Statement:** All relevant data are within the paper and its Supporting Information files.

## Abstract

### Background

The comprehensive effectiveness of the HPV vaccine has been widely acknowledged. However, challenges such as dosing adherence and limited budgets have led to delays in HPV vaccination implementation in many countries. A potential solution to these issues could lie in a one-dose vaccination with an HPV vaccine, as indicated by promising outcomes in multiple studies.

### Methods

In this systematic review and meta-analysis, we examine the comparative effectiveness of the one-dose vaccination with an HPV vaccine against two- and three-dose regimens. Our investigation focuses on clinical efficacy, encompassing the prevention of HPV16, HPV18, and hrHPV infections, HSIL or ASC-H incidence, and CIN2/3 incidence.

### Results

Our analysis suggests that a single-dose HPV vaccine may offer effectiveness on par with two- or three-dose schedules. This conclusion is drawn from its capacity to confer immunogenic protection for at least 8 years of follow-up, coupled with its ability to mitigate infections and pre-cancerous occurrences.

### Conclusion

While our findings underscore the potential of the one-dose vaccination with an HPV vaccine, further research and prolonged study durations are necessary to establish robust evidence supporting this recommendation. As such, continued investigation will be critical for informing vaccination strategies

**Funding:** These findings are the result of work supported by Universitas Muhammadiyah Purwokerto. Indonesia. The views expressed in this paper are those of the authors, and no official endorsement by Universitas Muhammadiyah Purwokerto is intended or should be inferred. NAN and MJP received no financial compensation for their contributions to this work. The funders had no role in study design, data collection and analysis, decision to publish, or preparation of the manuscript.

**Competing interests:** The authors have declared that no competing interests exist.

## Introduction

Cervical cancer stands as the fourth most prevalent cancer among women worldwide and is predominantly linked to Human Papillomavirus (HPV) infection [1]. This infection has been recognized to be responsible for about a quarter-million death-related to cervical cancer every year. Among the known HPV types, it is observed that there exist 20 HPV variants that are more prevalent in women with cervical cancer compared to those with normal cytology. Notably, HPV types 16 and 18 are categorized as carcinogenic or high-risk HPV (hrHPV), and they are particularly prominent, causing about 70% of all cases of cervical cancer. Conversely, there are additional HPV types (such as type 6 and 11) that fall under the classification of non-carcinogenic or low-risk HPV (lrHPV) [2]. Therefore, prevention of hrHPV infection will, as a matter of course, protect against cervical cancer.

Since the nature of cervical cancer development has almost been understood perfectly [3], the prevention strategies of cervical cancer have been introduced in several countries, considering HPV vaccination and cervical screening as primary and secondary prevention, respectively [4]. Previous studies showed that HPV vaccines are considerably effective in stimulating the antibody-specific hrHPV and potentially prevent cervical cancer in the future [5].

Three commercial HPV vaccines are currently available in the market, each vaccine offers not only protection to HPV16 and HPV18 infection, as the leading cause of cervical cancer, but also some additional clinical benefits [6–8]. Five types of HPV vaccines are currently available through private purchasing in the Chinese market: Cecolin⑬ (domestic HPV-2), Cervarix⑬ (imported HPV-2), Gardasil⑬ (HPV-4), Gardasil⑬9 (HPV-9), and a new domestically produced bivalent vaccine (Walrinvax™, Recombinant Human Papillomavirus Bivalent [Types 16, 18] [9]. While quadrivalent vaccine also offers protection for HPV6 and HPV11 as the leading cause of genital warts, bivalent and nonavalent vaccines also provide additional protection to other hrHPV including HPV31, HPV33, HPV45, HPV52, and HPV58 as the additional cause of cervical cancer with different degrees of protection to each type of virus [10].

During its initial introduction, HPV vaccines were suggested for three doses of administration to achieve some degree of protection against HPV infection and cervical cancer new cases [11]. During its application, also confirmed with continuing clinical trials, the immunogenicity profile of two doses of HPV vaccine was accidentally comparable with three doses of application [12]. These findings suggested a substantial reduction in cost-related vaccination since it has become one of the leading implementation barriers in several countries. The latest scattered findings from various studies showed that one dose of administration could potentially provide comparable results with two or even three doses of administration.

Several studies showed that the implementation of HPV vaccination as a cervical cancer prevention policy faces several issues such as coverage, acceptance, and the high price of the vaccine [13]. The reduction of cost-related vaccines will substantially influence the decision regarding implementing the policy in a country. Therefore, this study will initially start with a systematic review and meta-analysis that mainly focus on extracting information regarding the immunogenicity of the HPV vaccine that has been administered, both intentionally or unintentionally, for one dose compared to two or three doses of administration.

## Methods

We systematically searched for all studies that administered one dose of HPV vaccine intentionally or unintentionally from two electronic databases (PubMed and EMBASE). Two keywords ('HPV vaccine' AND 'one dose' "One dose"[tw] OR "Single dose"[tw] OR "first dose"[tw] OR "initial dose" [tw]) and their respective MeSH term and text word were used to identify all related studies in the PubMed database. These similar keywords were also used in

EMBASE database using exp (explosion search), ab (abstract), and ti (article title) commands. The last searching process was done on November 18th, 2022. in order to expand the search result, a snowball search was done by identifying the reference of both included studies and existing review articles. For risk of bias (RoB) assessment, Risk of Bias assessment tool version 2 and ROBINS-1 assessment tool for RCTs and pbservational studies, consecutively.

## Data collection and analysis

Since clinical outcomes are substantial in clinical decision-making, we only included studies that provide measurable clinical data. The results in this study was focused on the prevention of infection (both all hrHPV or HPV16 and HPV18 only) and prevention of pre-cancer/cancer development (HSIL or ASC-H and CIN2/3). Another inclusion criterion for this systematic review and meta-analysis is that the article is written in English. Two reviewers (DS and NAN) screened and assessed the detected studies independently, and any disagreement was discussed and solved with a third reviewer (MJP). Based on the Preferred Reporting Items for Systematic Reviews and Meta-Analyses (PRISMA) statement, we will extract the following data: authors, year published, country, study design, vaccine types, study population, number of subjects in each group, follow up time, outcomes and main finding of the included studies.

## Meta-analysis

To perform a meta-analysis, we extracted the incidence of HPV infection and pre-cervical cancers from both intervention groups (1 dose and the combination of two and or three doses). The HPV infection is differentiated into the infection caused by (1) HPV16 and HPV18 only and (2) all types of high-risk HPV (hrHPV). Moreover, other clinical outcomes in this meta-analysis have investigated the impact of one-dose vaccination with an HPV vaccine compared to 2 or 3 doses on (1) HSIL or ASC-H and (2) CIN2/3.

Risk Ratio (RR) was calculated from the number of events (infections and pre-cancers) in one-dose vaccination with an HPV vaccine groups compared to the groups who received two and or three doses of HPV vaccines using random-effects models to obtain the vaccine efficacy and or effectiveness. To deal with heterogeneity that could cause by the differences in sample characteristics and its methods, we estimated a heterogeneity test by calculating the $I^2$ score according to Cochrane Q test results. This method presents a value of heterogeneity ranging from 0% to 100%. According to Cochrane's recommendation, $I^2$ of 50% or higher is considered substantial heterogeneity. This meta-analysis will be performed using RevMan 5.3.

# Results

## Screening flow

From each database, this study found 405 and 494 from PubMed and Embase, respectively. After duplication removal (253 articles), there were 566 articles were excluded during abstract and title screening with the following reasons: studies are not about a single dose HPV vaccines (475 articles), communication articles including commentaries and letter to the editors (N = 18), articles are not written in English (N = 9), review articles (N = 47), qualitative and case report article (N = 3), official report documents (N = 5), and clinical guideline and recommendation (N = 9). During the full text screening, there were 64 out of 80 articles were excluded due to various reasons and 7 articles were included during snowball searching (Fig 1). Finally, 23 articles were included in the qualitative review and finally, there were only 11 articles were included in the meta-analysis since they provide explicit quantitative data on HPV vaccine effectiveness.

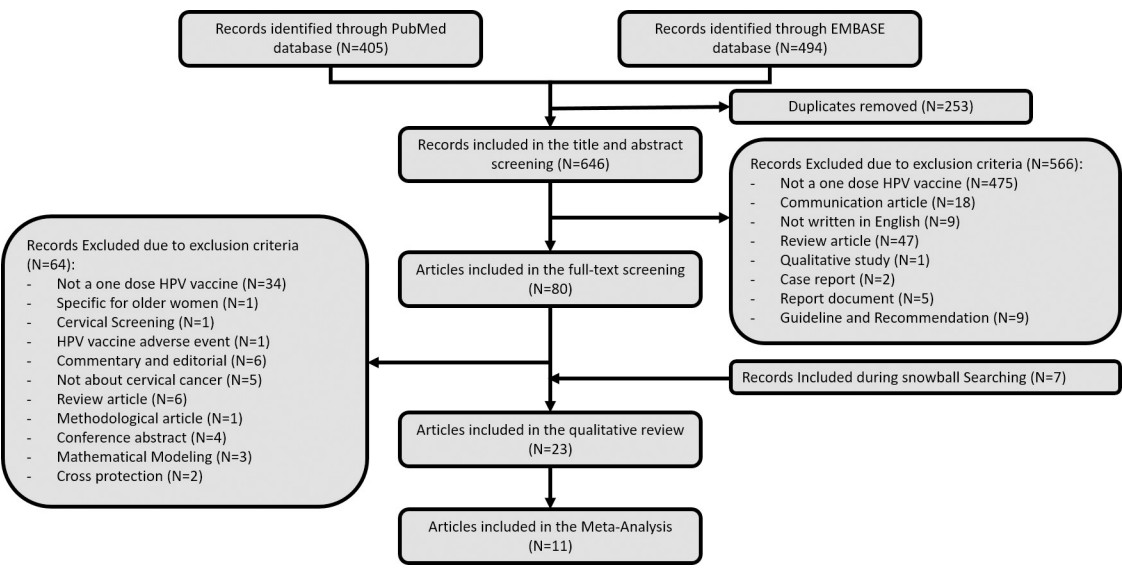

**Fig 1. Screening process.**

## Risk of bias assessment

The result of quality assessment for RCTs and observational studies is shown in Figs 2 and 3. The risk of bias assessment using RoB 2 showed majorities of the studies (1–5) [14–18] presented a sufficient description of the randomization process. Meanwhile, three trials showed a high risk of bias due to deviations from the intended interventions. The outcomes of all the trials were estimated sufficiently and led the low bias by missing outcomes data. Five studies(1–4,6) [14–17, 19] picted that the method of measuring the outcome was not inappropriate, and the outcome measurement did not differ between intervention groups. Some concerns in the selection of the reported results was shown in two studies(1,5) [14,18]. The overall result of the quality assessment showed that the four studies(1,5,7) [14, 18, 20] had some concern of the risk of bias, and the remaining studies(2–4,6) [15–17, 19] demonstrated a high risk of bias.

The assessment results using ROBINS-i showed that there is a low risk of bias in the studies by presenting adequate reporting (Fig 3). However, some studies(8–14) had a serious concern on the bias due to the confounding. Most of the studies exhibited sufficient reporting on the selection of participants, classification of interventions, and low of risk of bias due to deviations from intended interventions. A high selection bias that arises due to exclusion of individuals with missing information about intervention status was found in four studies (10,13,15,16). Finally, the measurement of outcomes and selection of the reported result were sufficiently reported.

## Study characteristics

The included studies were performed from various countries, including Australia [21–24], India [25–27], Canada [28], Fiji [29], United States [30, 31], Denmark [32], Costa Rica [17–20], Scotland [33, 34], Uganda [35], Worldwide [16], the Netherlands [36] and Tanzania [14, 15]. Two updated studies, one from India [26] and another one from Costa Rica [17], were included in the review but not in Meta-Analysis due to double counting concerns. Most included studies using observational cohort (11 studies) [21, 23–27, 29, 31–34, 37] and Randomized Control Clinical Trials (RCTs) (6 studies) [14–19]. Furthermore, both Quadrivalent

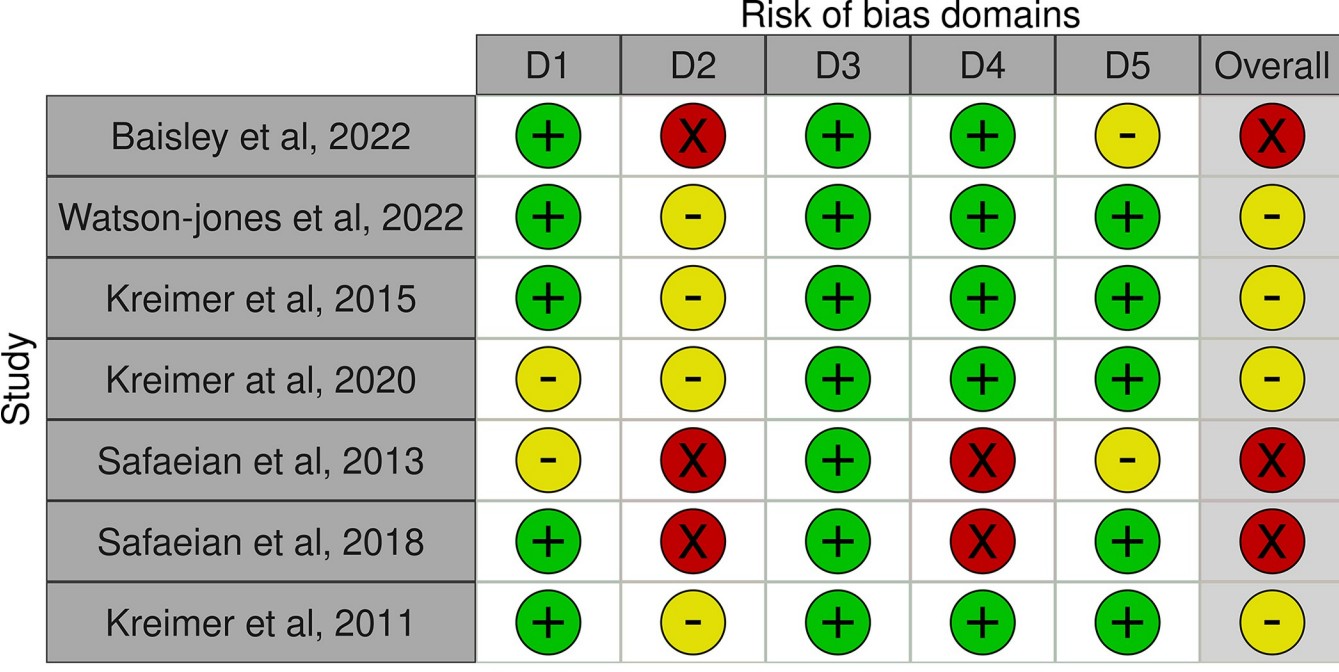

**Fig 2. Risk of bias using RoB (risk of bias) version 2 for RCTs studies.**

[21–29, 31, 32, 37] and Bivalent [16–20, 33–36] vaccines were almost equally studied. In addition, study comparing exclusively Bivalent versus Nonavalent [15] and Bivalent, Quadrivalent and Nonavalent [14] were also included in this review. There is a considerably wide variety of study population age (9 to 29 years old), the investigated group arms (unvaccinated to a complete three doses), duration of follow up (1 to 11 years long). Additionally, various clinical outcomes, both intermediate and endpoint, were investigated, including immunogenicity, infection prevention, and the prevention of pre-cancer and cancer itself.

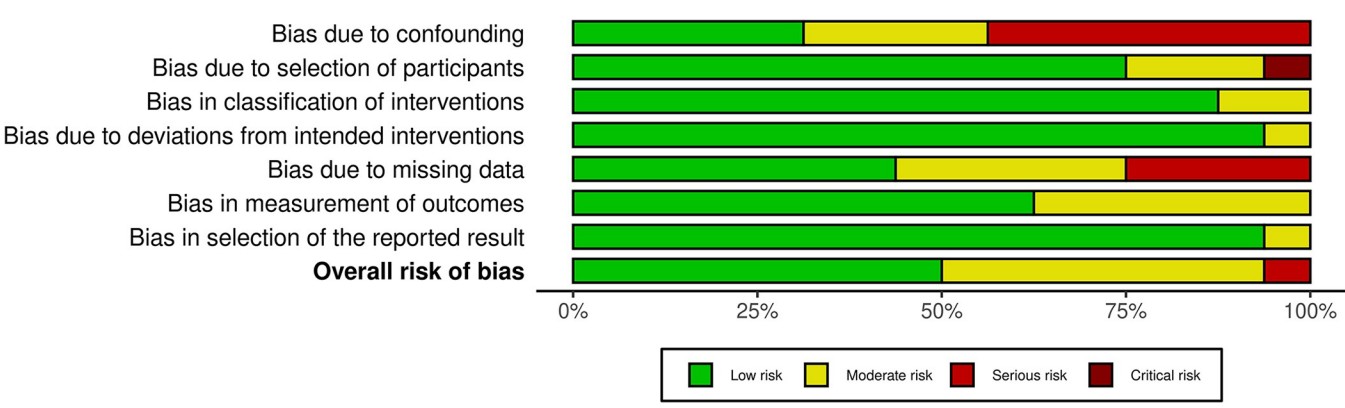

**Fig 3. Risk of bias assessment using ROBINS-I for observational studies.**

Among 23 included studies, more than half of them conclusively mention that one-dose vaccination with an HPV vaccine provides similar vaccine effectiveness compared to two or three-doses of vaccination [14–16, 19, 20, 25–29, 32, 33, 35, 38]. In addition, several studies suggest that antibody is available to protect the women who received one-dose vaccination with an HPV vaccine for at least 4 to 8 years, and immune memory was induced in that particular group [27, 39–41]. However, three studies mentioned that the one-dose vaccination with an HPV vaccine did not significantly reduce the incidence of both pre-cancer and infection rates compared to unvaccinated group [21, 22, 33]. Two studies conclude that although one-dose vaccination with an HPV vaccine is immunogenic and reduces the incidence of pre-cancers, however, the protection of this particular dose is not equal to two or three doses [23, 36] (Table 1).

## The effectiveness of HPV vaccine against HPV16 and HPV18 infection

Currently, most of the evidence shows that the prevalence of HPV16 and HPV18 infection in cervical cancer patients is the highest among other serotypes of HPV. Therefore, HPV16 and HPV18 infection incidence is closely related to cervical cancer cases. Six included studies describe that the risk of HPV16 and HPV18 infection among the one-dose group is slightly higher than the more-doses group (RR = 1.55, CI95% 1.18–2.04, *p value* 0.002) (Fig 4). this result is influenced mainly by a study from India [27] and Scotland [34] (weight 29.3% and 28.2%) where the study results were possibly underestimated the potential effectiveness among younger girls. furthermore, the comparison between one- and two doses of HPV vaccine also shows a differences in HPV16 and HPV18 infection rate between two groups (RR = 1.18, CI95% 0.92–1.44, *p-value* 0.10) (Fig 5).

## The effectiveness of HPV vaccine against hrHPV infection

In addition to HPV types 16 and 18, which collectively contribute to 70% of all cervical cancer cases, a broader spectrum of high-risk HPV types significantly impacts cervical cancer incidence. Notably, HPV types 16, 18, 31, 33, 45, 52, and 58 collectively account for approximately 90% of all cervical cancer cases. This comprehensive coverage of high-risk HPV types underscores the importance of vaccines such as the 9-valent HPV vaccine, which specifically addresses these strains to mitigate cervical cancer risk [42, 43]. This study also explores the vaccine effectiveness in preventing the infection of hrHPV among vaccinated girls. This meta-analysis shows that the effectiveness of one-dose vaccination with an HPV vaccine on preventing hrHPV infection are slightly lower than more-doses (RR = 1.27, CI95% 1.02–1.57, *p value* 0.03) (Fig 6) and two doses alone (RR = 1.14, CI95% 1.03–1.27, *p value* 0.01) (Fig 7).

## The effectiveness of HPV vaccine on preventing HSIL or ASC-H incidence

In the pap smear test, although Atypical Squamous Cell-H (ASC-H) is not included as a cancer cell, it could be part of High-Grade Squamous Intraepithelial Lesion (HSIL) which is included in a pre-cancer category. Therefore, a further confirmation test is required since HSIL may become cervical cancer if not treated quickly. In this study, the impact of one-dose vaccination with an HPV vaccine on preventing the incidence HSIL or ASC-H is apparently similar with more-doses (RR 1.29; 95% CI 0.88–1.88; *p-value* 0.19) (Fig 8) or two-doses alone (RR 1.01; 95% CI 0.74–1.37; *p-value* 0.97) (Fig 9). Although this result was only generated from a few studies, the individual studies showed almost comparable results and patterns, especially comparing one dose versus more doses. However, these analyses found heterogeneity among supported studies ($I^2$ of 88% and 51%).

**Table 1. The characteristic of included studies.**

| No | Author, Year | Location | Study design | vaccine types | Study population | Number of population on each group | Study period | Outcomes | Main findings |
|---|---|---|---|---|---|---|---|---|---|
| 1 | Gertig, 2013 | Australia | Retrospective cohort study using linked data from registries | QV | 12–13 y.o | Unvaccinated—15192; 1 dose = 2568; 2 doses = 3412; 3 doses = 21199 | 5 years | Cervical Abnormalities (Histological and Cytological) | there was no significant reduction among those partially vaccinated (1 or 2 doses) compared to unvaccinated group |
| 2 | Crowe, 2014 | Australia | Case Control Analysis | QV | 11–27 y.o. | Unvaccinated—53761; 1 dose = 9649; 2 doses = 10950; 3 doses = 23106 | 3 years | CIN 2 or 3/ adenocarcinoma–in–situ (AIS)/ cancer | there was no statistically significance effectiveness of one dose HPV vaccine for high grade cervical abnormalities compared to unvaccinated group |
| 3 | Brotherton, 2015 | Australia | retrospective cohort study | QV | 26 or younger | 1 dose = 20,659; 2 doses = 27,500; 3 doses = 108,264 | 2.89 years (average) | CIN2, CIN3, AIS or mixed CIN3/ AIS | Any number of doses (1, 2 or 3) was found to be associated with lower rates of high grade and low grade cytology diagnoses |
| 4 | Sankaranarayanan, 2015 | India | cohort study | QV | 10–18 y.o. | 1 dose = 4950; 2 doses (1 and 60) = 3452; 2 doses (1 and 180) = 4979; 3 doses = 4348 | 60 months | Immunogenicity, incidence and prevalence | The short- term protection afforded by one dose of HPV vaccine against persistent infection with HPV 16, 18, 6, and 11 is similar to that afforded by two or three doses of vaccine and merits further assessment. |
| 5 | Kim, 2016 | Canada | Nested Case-Control | QV | 10–11 y.o. | Unvaccinated—5712; 1 dose = 327; 2 doses = 490; 3 doses = 3675 | 8 years | Abnormal cytology | The adjusted odds of high-grade cervical abnormalities with at least 1 dose of vaccination compared with no vaccination were lower |
| 6 | Toh, 2016 | Fiji | Prospective cohort study | QV | 15–19 y.o. | Unvaccinated = 32; 1 dose = 40; 2 doses = 59; 3 doses = 66 | 6 years | Seropositivity | A single dose of 4vHPV elicits antibodies that persisted for at least 6 years, and induced immune memory, suggesting possible protection against HPV vaccine types after a single dose of 4vHPV |

*(Continued)*

**Table 1.** (Continued)

| No | Author, Year | Location | Study design | vaccine types | Study population | Number of population on each group | Study period | Outcomes | Main findings |
|---|---|---|---|---|---|---|---|---|---|
| 7 | Sankaranarayanan, 2018 | India | cohort study | QV | 10–18 y.o. | 1 dose = 4950; 2 doses (1, 60 days) = 3452; 2 doses (1, 180 days) = 4979; 3 doses = 4348 | 7 years | Immunogenicity, incidence and prevalence of HPV infection | in our observational cohort study, one dose HPV vaccine is immunogenic and provides protection against HPV 16 and 18 infections similar to the two- and three-dose vaccine schedules |
| 8 | Brotherton, 2019 | Australia | Retrospective cohort study | QV | under 15 y.o. | Unvaccinated = 48,845, 1 dose = 8,618; 2 doses = 18,190; 3 doses = 174,995 | 7 years | CIN 2 or 3/ adenocarcinoma–in–situ (AIS)/ cancer | One dose had comparable effectiveness as two or three doses in preventing high–grade disease in a high coverage setting |
| 9 | Markowitz, 2019 | US | Retrospective cohort study | QV | 20–29 y.o. | Unvaccinated = 1052; 1 dose = 303; 2 doses = 304; 3 doses = 2610 | 73, 58, and 76 months | HPV infection prevalence | Among women who received their first dose at age ≤18, estimated HPV vaccine effectiveness was high regardless of number of doses. |
| 10 | Verdoodt, 2019 | Denmark | Nationwide Cohort study | QV | less than 16 y.o. | Unvaccinated = 373,327; 1 dose = 10,480; 2 doses = 30,259; 3 doses = 174,532 | 8 years | CIN3+ and CIN2 or worse | We find substantial effectiveness of qHPV vaccination against high-grade cervical precancerous lesions, among women vaccinated with 1, 2, or 3 doses at ≤16 years of age. One-dose vaccination appeared to provide similar protection as 3-dose vaccination. |
| 11 | Rodriguez, 2020 | US | Retrospective cohort study | QV | 9–26 y.o. | Unvaccinated = 66,541; 1 dose = 13,630; 2 doses = 14,088; > = 3 doses = 38,823 | 1 year (at least) | CIN II, CIN III, and HSIL or ASC-H | Overall, our findings showed a similar degree of association between varying doses of 4vHPV vaccines and preinvasive cervical lesions among adolescents who received the vaccine between the ages of 15 and 19 years |

*(Continued)*

**Table 1.** (Continued)

| No | Author, Year | Location | Study design | vaccine types | Study population | Number of population on each group | Study period | Outcomes | Main findings |
|---|---|---|---|---|---|---|---|---|---|
| 12 | Basu P, 2021 | India | Prospective, cohort study | QV | 10–18 y.o | Unvaccinated = 1541; 1 dose = 4949; 2 doses = 4980; 3 doses = 4348 | 9 years | CIN2+; Incidence HPV 16/18 infections; HPV 16/18-related CIN2+ lesions | A similar incident of infection was observed across the vaccinated groups. Similar frequencies of persistent infection was showed among the dose groups. The adjusted efficacy of a single dose against persistent HPV 16/18 infection was 95,4% (85,0–99,9) The efficacy against persistent infection from all HPV types was 35,4% (3,7–56,0) |
| 13 | Kreimer, 2011 | Costa Rica | Non-Random Clinical Trial | BV | 18–25 y.o. | 1 dose = 196; 2 doses = 422; 3 doses = 2957 | 1 year | incidence and prevalence | Two doses of the HPV16/18 vaccine, and maybe even one dose, are as protective as three doses. |
| 14 | Safaeian, 2013 | Costa Rica | RCT | BV | 18–25 y.o. | 1 dose = 78; 2 doses (0/1 mo.) = 140; 2 doses (1/6 mo.) = 52; 3 doses = 120 | 4 years | Seropositivity | a single HPV16/18 vaccine dose induces an antibody response that was readily detected in all vaccinated young women at end of the 4-year follow-up, although the titers were lower than after two or three doses and the number of one and two dose recipients was limited. Results raise the possibility that even a single dose of HPV VLPs will induce long-term protection. |
| 15 | Kavanagh, 2014 | Scotland | Retrospective Cohort Studies | BV | 12–13 y.o | Unvaccinated—1018; 1 dose = 55; 2 doses = 106; 3 doses = 1100 | 8 years | Infection rate | Compared to unvaccinated group, Although one dose also showed a reduction in HPV type 16 and 18, (OR1⁄40.88, 95% CI 0.48, 1.6) this was not statistically significant (P 1⁄4 0.68) |

(*Continued*)

**Table 1.** (Continued)

| No | Author, Year | Location | Study design | vaccine types | Study population | Number of population on each group | Study period | Outcomes | Main findings |
|---|---|---|---|---|---|---|---|---|---|
| 16 | LaMontagne, 2014 | Uganda | Cross-sectional Study | BV | 10 y.o. | 1 dose = 36; 2 doses = 145; 3 doses = 195 | 33 months | GMTs | Even though immunogenicity with less than three doses did not meet a priori non-inferiority thresholds, antibody levels measured ≥24 months after last dose were similar to those of adult women who have been followed for more than eight years for efficacy. |
| 17 | Kreimer, 2015 | Worldwide | RCT | BV | 15–25 y.o. | 1 dose = 292; 2 doses = 611; 3 doses = 11,104 | 4 years | Incident of infection | 4 years after vaccination of women aged 15–25 years, one and two doses of the HPV-16/18 vaccine seem to protect against cervical HPV-16/18 infections, similar to the protection provided by the three-dose schedule |
| 18 | Cuschieri, 2016 | Scotland | Retrospective Cohort Studies | BV | 12–13 y.o | Unvaccinated—3619; 1 dose = 177; 2 doses = 300; 3 doses = 1853 | 10 years | Infection rate | We demonstrate the potential effectiveness of even one dose of HPV vaccine on vaccine-type infection. |
| 19 | Safaeian, 2018 | Costa Rica | RCT | BV | 18–25 y.o. | 1 dose = 134; 2 doses (0 and 1 mo) = 193; 2 doses (0 and 6 mo) = 79; 3 doses = 2,043 | 7 years | infection rate, seropositivity | A low prevalence of HPV16/18 infections was observed for all dose groups, suggesting that the protection afforded by even a single dose may be long lived. |
| 20 | Pasmans, 2019 | The Netherlands | Cross sectional study | BV | 12 y.o. | Unvaccinated = 51; 1 dose = 239; 2 doses = 222; 3 doses = 378 | 7 years | HPV-type-specific IgG and IgA-antibody levels, IgG-isotypes and avidity indexes | One-dose of the 2vHPV vaccine is immunogenic, but results in less B- and T-cell memory and considerable lower antibody responses when compared with more doses |

(*Continued*)

**Table 1.** (Continued)

| No | Author, Year | Location | Study design | vaccine types | Study population | Number of population on each group | Study period | Outcomes | Main findings |
|---|---|---|---|---|---|---|---|---|---|
| 21 | Kreimer, 2020 | Costa Rica | RCT | BV | 18–25 y.o. | 1 dose = 112; 2 doses = 62; 3 doses = 1365 | 11 years | HPV infection | More than a decade after HPV vaccination, single-dose VE against HPV16 or 18 infection remained high and HPV16 or 18 antibodies remained stable. A single dose of bivalent HPV vaccinemay induce sufficiently durable protection that obviates the need formore doses |
| 22 | Watson-jones D, 2022 | Tanzania | RCT, non-inferiotity trial | BV and 9-valent | 9–14 years | 1 dose BV = 155; 2 doses BV = 155; 3 doses BV = 155; 1 dose 9-valent = 155; 2 doses 9-valent = 155; 3 doses 9-valent = 155 | 24 months | Seropositivity Safety | Non-inferiority of HPV 16 seroconversion at 24 months was met for one dose compared with two or three doses for both vaccines. Participants in one dose arm of both vaccines were anti = -HPV 18 antibody positive at 24 minths. HPV 16 and 18 antibody GMCs were higher among girls receiving two and three doses than receiving single dose. The GM antibody index across group was not difference significantly. The serious adverse events (SAEs) also was not meaningfully difference among arms. |
| 23 | Baisley K, 2022 | Tanzania | RCT, immunobridging analysis | BV, QV, and 9-valent | 9–14 | DoRIS 1 dose BV = 154; 1 dose 9-valent = 155 CVT 1 dose BV = 115 IARC 1 dose QV = 139 | 24 months | GMCs | HPV 16 and 18 antibody GMCs were higher after one dose of the BV in DoRIS than CVT (the difference was not significant). Single dose of HPV vaccine induces immune responses that are comparable in different populations, and is likely to be effective against persistent HPV 16 and HPV 18 infection |

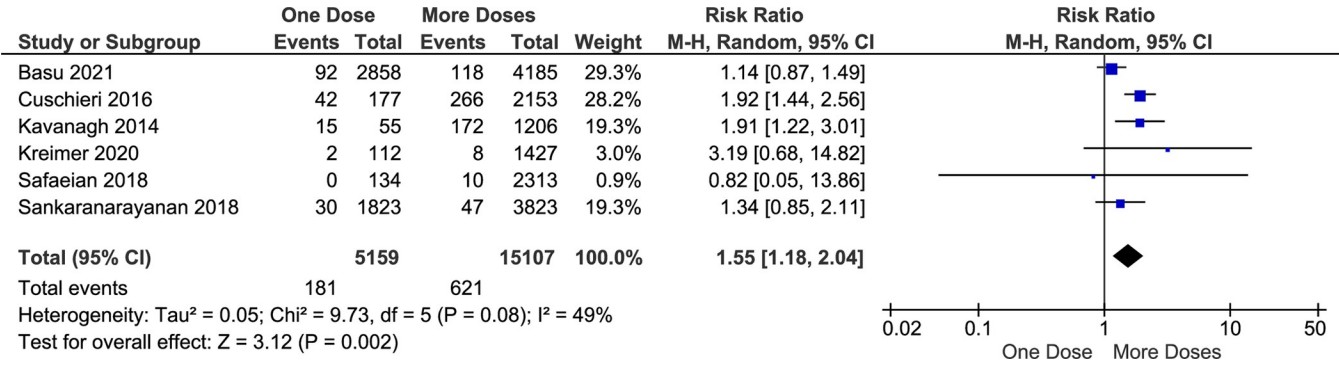

**Fig 4. The effectiveness of one- and more-doses HPV vaccine on preventing HPV16 and HPV18 infection.**

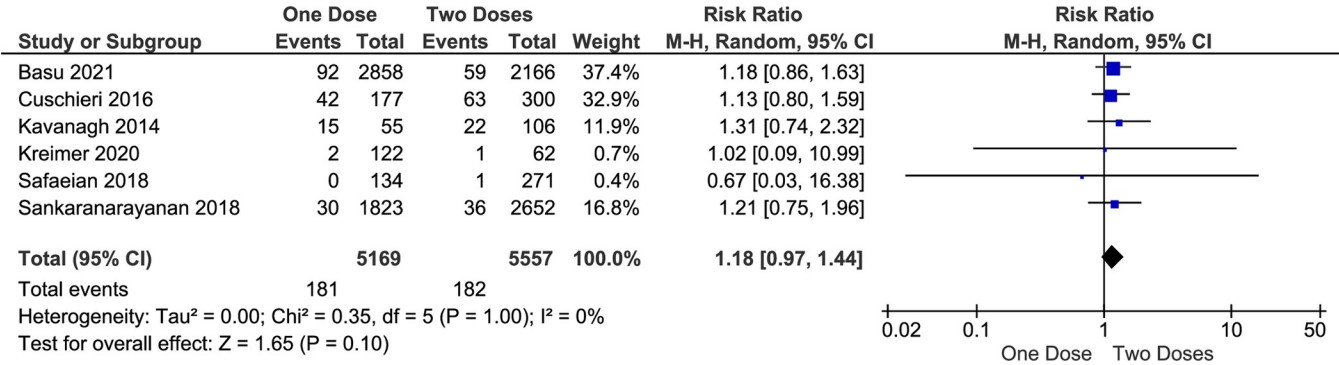

**Fig 5. The effectiveness of one- and two-doses HPV vaccine on preventing HPV16 and HPV18 infection.**

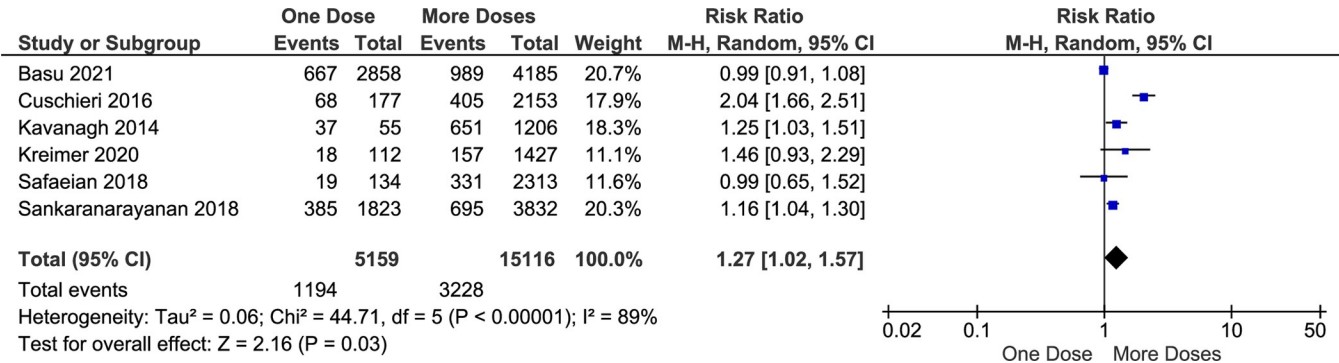

**Fig 6. The effectiveness of one- and more-doses HPV vaccine on preventing hrHPV infection.**

## The effectiveness of HPV vaccine on preventing CIN2/3 incidence

Cervical Intraepithelial Neoplasia (CIN) are precancerous stages where some abnormal cell grows on the surface of the cervix, and it can be found during cervical screening, for example, pap smear test and Visual Inspection using Acetic Acid (VIA). The classification of CIN is

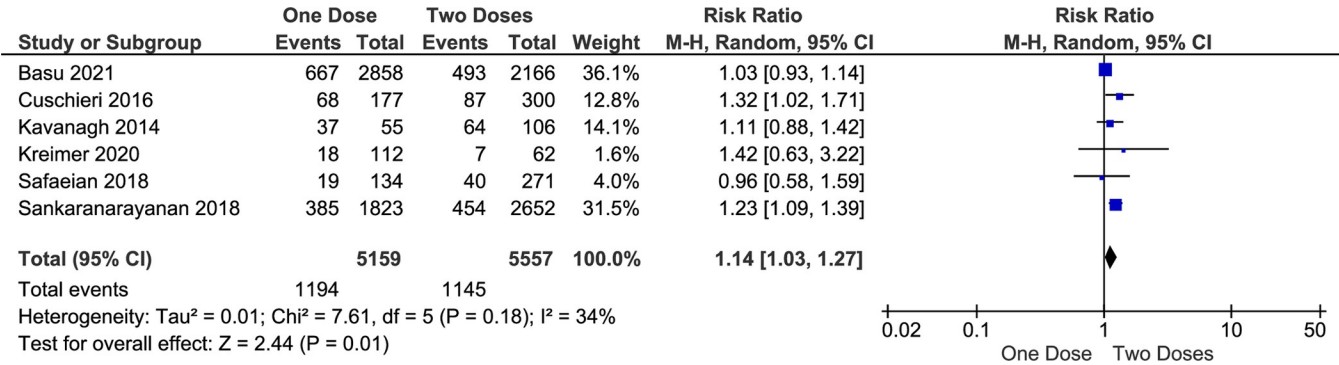

**Fig 7. The effectiveness of one- and two-doses HPV vaccine on preventing hrHPV infection.**

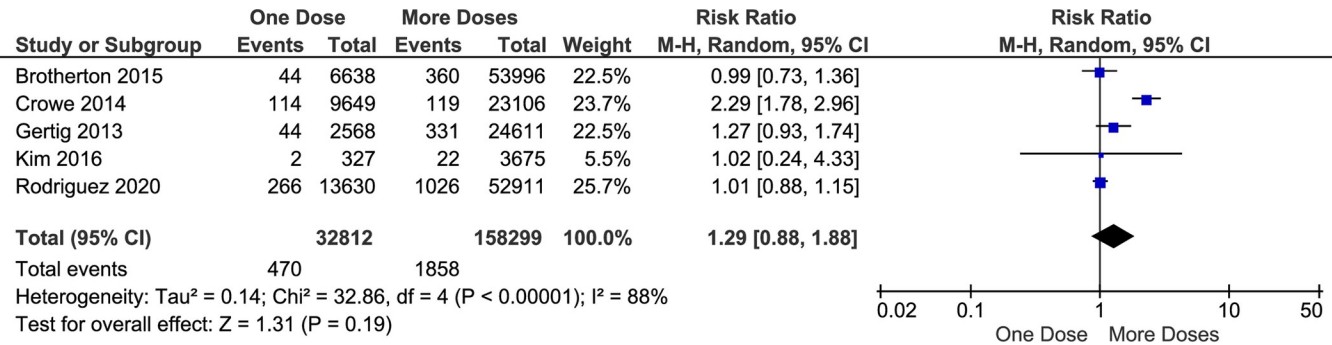

**Fig 8. The effectiveness of one- and more-doses HPV vaccine on preventing HSIL or ASC-H incidence.**

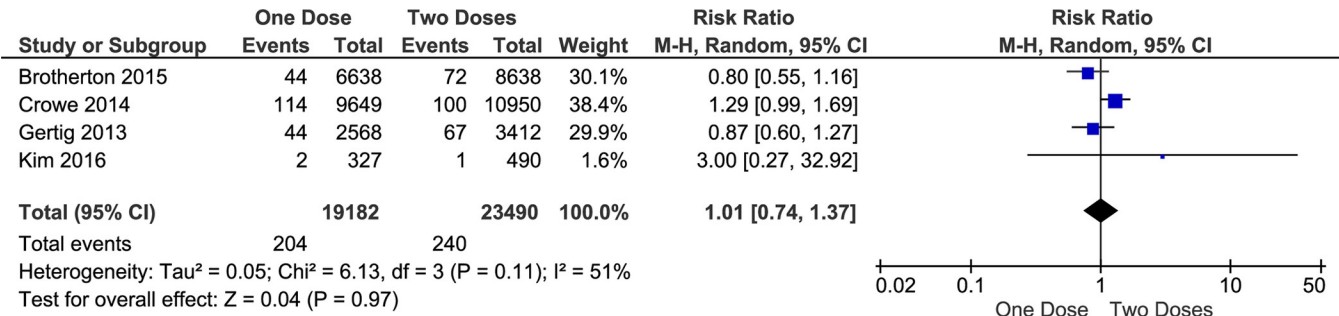

**Fig 9. The effectiveness of one- and two-doses HPV vaccine on preventing HSIL or ASC-H incidence.**

determined according to the proportion of the cervix, particularly the epithelial surface. CIN 2 usually describes that dysplasia affects about one-third to two-thirds of the epithelium. While CIN 3 represents that more than two-thirds of the epithelial layer of the cervix is changed and it is considered the most severe form of CIN. This meta-analysis finds that women who received one-dose vaccination with an HPV vaccine showed comparable effectiveness with women who received more doses to prevent CIN2/3 (RR = 1.54, CI95% 0.91–2.62, *p value* 0.11). However, there is high heterogeneity among studies in this analysis ($I^2$ of 94%) (Fig 10).

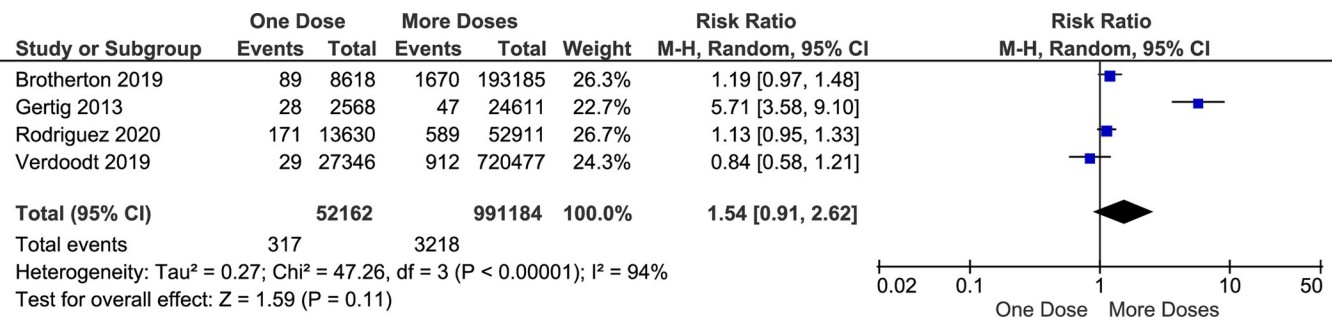

**Fig 10. The effectiveness of one- and more-doses HPV vaccine on preventing CIN2/3 incidence.**

## Discussion

The nationwide implementation of HPV vaccination in a country often faces problems, including the low adherence to second or third dose and limited budget to provide complete doses of the HPV vaccine, especially in low- and middle-income countries. A one-dose vaccination with an HPV vaccine could solve these issues significantly reduce those problems. However, valid and quantitative evidence on the efficacy or effectiveness of one-dose vaccination with an HPV vaccine should be available beforehand to help the decision-making process. This systematic review focus on the effectiveness of the one-dose vaccination with an HPV vaccine compared to more doses (two- and three-doses) on protecting infection and pre-cancer incidence. From this systematic review and meta-analysis, we found that most of the included studies support that one-dose vaccination with an HPV vaccine induces HPV-specific antibodies up to 8 years, providing immunogenic memory and as effective as two- or more doses of vaccination on preventing infection and pre-cancer condition. However, few studies explain that the antibodies are not equal to two- or three doses, and the reduction in infection rate and the incidence of pre-cancer is not significant. Therefore, more studies and more extended study periods are required to provide a definitive conclusion of the effectiveness of the one dose HPV vaccine.

This study strengthens the previous systematic review on one-dose vaccination with an HPV vaccine [44] since this study includes not only Randomized Controlled clinical Trials but also observational studies that have a longer duration of follow-up. In addition, this review also considers the end-point clinical outcomes, including pre-cancer and cancer incidence, which are more critical in the perspective of patients, clinicians, and other decision-makers. Ultimately, this meta-analysis substantially provides a quantitative pooled analysis of relative risk between one-dose versus more doses and two-doses of HPV vaccines across available studies in the world. In addition, this review does not support the previous review from Markowitz et al., which clearly explained that the immunogenicity of the one-dose vaccination with an HPV vaccine is inferior to three doses.

This study extracted pooled quantitative vaccine effectiveness data on preventing both infection and pre cancers incidence. One of the exciting findings in this meta-analysis is that the effectiveness of a one-dose vaccination with an HPV vaccine is generally comparable with more doses in pre-cancer prevention but not on infection prevention. Several studies, mainly from Scotland and India, do not support the hypothesis of similar effectiveness between one- versus more-doses HPV vaccines on preventing hrHPV infection. Both studies explain that one-dose vaccination with an HPV vaccine does not sufficiently induce cross-protection, particularly on HPV31, HPV 33, and HPV 45. These studies

could be the main drive on why one-dose vaccination with an HPV vaccine does not as effective as more doses on preventing hrHPV infection.

Since several countries have implemented a two-dose HPV vaccination schedule, this review also evaluates the head-to-head comparison between one- versus two-dose HPV vaccines. According to our meta-analysis, the one-dose vaccination with an HPV vaccine provides comparable effectiveness with the two-dose HPV vaccine on the prevention of not only HPV16 and HPV18 but also HSIL or ASC-H incidence. On the other hand, although comparable effectiveness between two alternatives could not be presented in terms of hrHPV infection prevention, two studies that did not support this particular comparison explain that one-dose vaccination with an HPV vaccine has comparable effectiveness with more doses (the combination of two- and three-doses). This phenomenon could cause by the limited number of the one-dose group or the number of incidences in the one-group arm.

One of the limitations of this study is that there were only 23 included studies and several of them are updated studies. Moreover, only 4–6 studies can be included in each meta-analysis. This condition restricts our study when sub-group analysis should be performed to evaluate the impact of one dose on a specific age group. Furthermore, this study did not assess the effectiveness of the one-dose vaccination with an HPV vaccine on the endpoint clinical outcome, which is cervical cancer incidence. However, although longer follow-up time is required before deciding on implementing a one-dose vaccination with an HPV vaccine schedule is made, these consistent findings provide a promising comparable benefits between one- and more-doses of HPV vaccines.

In conclusion, this study provides a comprehensive evaluation of the potential efficacy and feasibility of a one-dose vaccination with an HPV vaccine regimen as a solution to challenges in HPV vaccination implementation. By synthesizing evidence from diverse study types, the analysis demonstrates that a single dose of the HPV vaccine holds promise in offering comparable effectiveness to multi-dose regimens in preventing pre-cancerous conditions. Notably, while the vaccine's capacity to provide immunogenic protection for at least 8 years is encouraging, discrepancies in infection prevention outcomes highlight the need for further investigation. The inclusion of specific HPV types that influence vaccine effectiveness adds depth to the understanding of these dynamics.

The study's insights, while subject to limitations in terms of study numbers and sub-group analyses, contribute substantively to the discourse surrounding HPV vaccination strategies. The findings underscore the potential benefits of simplifying vaccination regimens, particularly in resource-constrained settings. However, the decision to implement a one-dose regimen should be informed by extended follow-up studies and rigorous clinical outcome assessments. Ultimately, this investigation points to a path of enhanced vaccination coverage and improved accessibility, with the overarching goal of mitigating cervical cancer's global burden. This study was not to mention all currently ongoing one-dose trials: KEN-SHE, DORIS, ESCUDDO, and probably others.

## Supporting information

**S1 Checklist. PRISMA 2020 checklist.**
(DOCX)

**S1 Data. Search result.**
(DOCX)

## Author Contributions

**Conceptualization:** Didik Setiawan, Maarten J. Postma.

**Data curation:** Nunuk Aries Nurulita.

**Formal analysis:** Nunuk Aries Nurulita.

**Investigation:** Nunuk Aries Nurulita.

**Methodology:** Nunuk Aries Nurulita.

**Project administration:** Sudewi Mukaromah Khoirunnisa.

**Resources:** Sudewi Mukaromah Khoirunnisa.

**Supervision:** Didik Setiawan, Maarten J. Postma.

**Validation:** Didik Setiawan.

**Visualization:** Didik Setiawan, Sudewi Mukaromah Khoirunnisa.

**Writing – original draft:** Didik Setiawan, Sudewi Mukaromah Khoirunnisa.

**Writing – review & editing:** Didik Setiawan, Maarten J. Postma.

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
