## [Decision Letter · Decision Letter 0]

24 Aug 2023

PONE-D-23-25664The Clinical Effectiveness of One-Dose HPV Vaccine: A meta-analysis of 902,368 vaccinated womenPLOS ONE

Dear Dr. Setiawan,

Thank you for submitting your manuscript to PLOS ONE. After careful consideration, we feel that it has merit but does not fully meet PLOS ONE’s publication criteria as it currently stands. Therefore, we invite you to submit a revised version of the manuscript that addresses the points raised during the review process.

We look forward to receiving your revised manuscript.

Kind regards,

Gregory Halle-Ekane, M.D;FWACS

Academic Editor

PLOS ONE

Journal Requirements:

"NO"

5. We note you have included a table to which you do not refer in the text of your manuscript. Please ensure that you refer to Table 1 in your text; if accepted, production will need this reference to link the reader to the Table.

Reviewers' comments:

Reviewer's Responses to Questions

**Comments to the Author**

1. Is the manuscript technically sound, and do the data support the conclusions?

Reviewer #1: Yes

Reviewer #2: Yes

2. Has the statistical analysis been performed appropriately and rigorously? 

Reviewer #1: Yes

Reviewer #2: Yes

3. Have the authors made all data underlying the findings in their manuscript fully available?

Reviewer #1: Yes

Reviewer #2: Yes

4. Is the manuscript presented in an intelligible fashion and written in standard English?

Reviewer #1: Yes

Reviewer #2: No

5. Review Comments to the Author

Reviewer #1: Setiawan et al. have addressed the challenges surrounding HPV vaccination implementation, where dosing adherence and budget constraints have posed significant hurdles. Their systematic review and meta-analysis delves into the potential of a one-dose HPV vaccine, supported by promising outcomes in multiple studies. By comparing the clinical efficacy of one-dose regimens against two- and three-dose schedules, the investigation focuses on preventing HPV infections, HSIL or ASC-H incidence, and CIN2/3 occurrences, encompassing HPV16, HPV18, and hrHPV.

The analysis indicates that a singular HPV vaccine dose could match the effectiveness of multi-dose regimens. Notably, the ability to provide immunogenic protection for at least 8 years, coupled with the capacity to mitigate infections and pre-cancerous conditions, is promising. However, while this study underscores the potential of the one-dose approach, further research and extended study periods are essential to substantiate these findings robustly. Continued investigation will be pivotal in shaping vaccination strategies and optimizing cervical cancer prevention efforts.

WHO now recommends one-dose regime of HPV-vaccine:

"The World Health Organization (WHO) has recommended shifting from a two-dose to one-dose vaccine regimen against the Human Papillomavirus (HPV) – something that could help expand vaccine coverage amongst millions of girls and young women in lower-income regions where HPV is most prevalent, as well as saving costs. "

https://healthpolicy-watch.news/who-updates-hpv-vaccine-schedule/

"The 4-7 April convening of the WHO Strategic Advisory Group of Experts on Immunization (SAGE) evaluated the evidence that has been emerging over past years that single-dose schedules provide comparable efficacy to the two or three-dose regimens.

SAGE’s review concluded that a single-dose Human Papillomavirus (HPV) vaccine delivers solid protection against HPV, the virus that causes cervical cancer, that is comparable to 2-dose schedules. This could be a game-changer for the prevention of the disease; seeing more doses of the life-saving jab reach more girls."

https://www.who.int/news/item/11-04-2022-one-dose-human-papillomavirus-(hpv)-vaccine-offers-solid-protection-against-cervical-cancer

"Emerging evidence suggests that just one dose of the HPV vaccine is as effective as two or three—and that has huge implications for the battle against cervical cancer."

https://www.nationalgeographic.com/science/article/cervical-cancer-hpv-vaccine-single-dose

Comments

1. To ensure precise and efficient peer reviews and streamline post-review editing procedures, it is advisable to implement a line numbering system for all submitted manuscripts.

2. Usually, the abstract is divided into Background, Material and Methods, Results and Conclusion:

"Background: The comprehensive effectiveness of the HPV vaccine has been widely acknowledged. However, challenges such as dosing adherence and limited budgets have led to delays in HPV vaccination implementation in many countries. A potential solution to these issues could lie in a one-dose HPV vaccine, as indicated by promising outcomes in multiple studies.

Methods: In this systematic review and meta-analysis, we examine the comparative effectiveness of the one-dose HPV vaccine against two- and three-dose regimens. Our investigation focuses on clinical efficacy, encompassing the prevention of HPV16, HPV18, and hrHPV infections, HSIL or ASC-H incidence, and CIN2/3 incidence.

Results: Our analysis suggests that a single-dose HPV vaccine may offer effectiveness on par with two- or three-dose schedules. This conclusion is drawn from its capacity to confer immunogenic protection for at least 8 years of follow-up, coupled with its ability to mitigate infections and pre-cancerous occurrences.

Conclusion: While our findings underscore the potential of the one-dose HPV vaccine, further research and prolonged study durations are necessary to establish robust evidence supporting this recommendation. As such, continued investigation will be critical for informing vaccination strategies."

3. Keywords: human papillomavirus, HPV vaccine, clinical efficacy, one-dose vaccination, single-dose regimen, cervical cancer prevention, immunogenic protection, pre-cancerous conditions, vaccination strategy"

4. Introduction, "Cervical cancer has becomes the second most common cancer globally and is mainly caused by

Human Papillomavirus (HPV) infection." => "Cervical cancer stands as the fourth most prevalent cancer among women worldwide and is predominantly linked to Human Papillomavirus (HPV) infection."

https://www.who.int/news-room/fact-sheets/detail/cervical-cancer

Sung H, Ferlay J, Siegel RL, Laversanne M, Soerjomataram I, Jemal A, et al. Global cancer statistics 2020: GLOBOCAN estimates of incidence and mortality worldwide for 36 cancers in 185 countries. CA Cancer J Clin. 2021:71:209–49. doi:10.3322/caac.21660.

5. Introduction, "Among the known HPV types. Some of HPV types (e.g. type 16 and 18) are categorized as carcinogenic HPV or high-risk HPV (hrHPV) while other types of HPV (e.g. type 6 and 11) are categorized as non-carcinogenic types

or low-risk HPV (lrHPV)" =>

"Among the known HPV types, it is observed that there exist 20 HPV variants that are more prevalent in women with cervical cancer compared to those with normal cytology. Notably, HPV types 16 and 18 are categorized as carcinogenic or high-risk HPV (hrHPV), and they are particularly prominent, causing about 70% of all cases of cervical cancer. Conversely, there are additional HPV types (such as type 6 and 11) that fall under the classification of non-carcinogenic or low-risk HPV (lrHPV)."

Arbyn M, Tommasino M, Depuydt C, Dillner J. Are 20 human papillomavirus types causing cervical cancer? J Pathol. 2014 Dec;234(4):431-5. doi: 10.1002/path.4424. PMID: 25124771.

6. This is an alternative version of the introduction:

"Cervical cancer ranks as the fourth most prevalent cancer among women globally and is predominantly attributed to Human Papillomavirus (HPV) infection[^1^]. This infection alone accounts for approximately a quarter-million deaths related to cervical cancer each year. Among the known HPV types, an observed disparity reveals the prevalence of 20 HPV variants in women with cervical cancer compared to those with normal cytology. Notably, HPV types 16 and 18, categorized as high-risk HPV (hrHPV), emerge as prominent culprits, responsible for about 70% of all cervical cancer cases. Conversely, distinct HPV types such as 6 and 11 fall into the classification of non-carcinogenic or low-risk HPV (lrHPV), underscoring the relevance of hrHPV infection prevention in safeguarding against cervical cancer.

As our understanding of cervical cancer's development reaches a mature phase[^3^], strategies for its prevention have taken shape globally, prioritizing HPV vaccination and cervical screening as primary and secondary preventive measures, respectively[^4^]. Previous investigations attest to the substantial efficacy of HPV vaccines in inciting hrHPV-specific antibody responses, holding the promise of future cervical cancer prevention[^5^].

In today's landscape, three commercial HPV vaccines grace the market, each offering protection not only against HPV16 and HPV18 infections, the primary culprits behind cervical cancer, but also presenting supplementary clinical benefits[^6^–^8^]. Beyond these, the quadrivalent vaccine addresses HPV6 and HPV11, leading agents of genital warts. Furthermore, bivalent and nonavalent vaccines extend their protective influence to encompass additional hrHPV strains, including HPV31, HPV33, HPV45, HPV52, and HPV58, each contributing to cervical cancer with varying degrees of impact[^9^].

In its nascent stages, HPV vaccines were recommended for three doses to confer an adequate degree of protection against HPV infections and the ensuing cervical cancer[^10^]. Subsequent application, corroborated by ongoing clinical trials, unveiled the surprising comparability of the immunogenicity profiles between two- and three-dose HPV vaccine administrations[^11^]. This revelation catalyzed discussions on substantial reductions in cost-associated vaccination barriers, a prevailing obstacle in numerous regions. Contemporary findings scattered across diverse studies further suggest that a single-dose administration potentially yields comparable outcomes to the traditional two- or three-dose regimens.

Amid the efforts to implement HPV vaccination as a cornerstone of cervical cancer prevention policies, concerns have arisen regarding coverage, acceptance, and the financial implications of the vaccine[^12^]. Alleviating cost-related barriers stands to significantly impact the decision-making process regarding policy implementation. In this context, this study embarks on a foundation of a systematic review and meta-analysis, primarily focused on extracting comprehensive insights into the immunogenicity of the HPV vaccine. Intentionally or unintentionally administered, the inquiry bridges one-dose applications against the backdrop of two- or three-dose protocols, with the overarching aim of contributing informed perspectives to the ongoing discourse.”

7. Results, "In addition to HPV16 and HPV18, there are several types of HPV that have been known as highrisk

HPV, including HPV31, HPV33, HPV45, HPV52, and HPV58. Those other types of HPV are included as hrHPV since they are also highly detected on cervical cancer patients" =>

"In addition to HPV types 16 and 18, which collectively contribute to 70% of all cervical cancer cases, a broader spectrum of high-risk HPV types significantly impacts cervical cancer incidence. Notably, HPV types 16, 18, 31, 33, 45, 52, and 58 collectively account for approximately 90% of all cervical cancer cases. This comprehensive coverage of high-risk HPV types underscores the importance of vaccines such as the 9-valent HPV vaccine, which specifically addresses these strains to mitigate cervical cancer risk."

8. Results, "The effectiveness of HPV vaccine on preventing CIN II/III incidence"

In the manuscript, there appears to be a mixture of terminologies related to cervical intraepithelial neoplasia (CIN) stages, particularly the use of both 'CIN II/III' and 'CIN 2/3'. To maintain consistency, I recommend that the authors conduct a thorough search and replace instances of 'CIN II/III' with 'CIN2/3' throughout the manuscript. This clarification will ensure uniformity and enhance the clarity of the manuscript's presentation.

9. Discussion, add a conclusion "In conclusion, this study provides a comprehensive evaluation of the potential efficacy and feasibility of a one-dose HPV vaccine regimen as a solution to challenges in HPV vaccination implementation. By synthesizing evidence from diverse study types, the analysis demonstrates that a single dose of the HPV vaccine holds promise in offering comparable effectiveness to multi-dose regimens in preventing pre-cancerous conditions. Notably, while the vaccine's capacity to provide immunogenic protection for at least 8 years is encouraging, discrepancies in infection prevention outcomes highlight the need for further investigation. The inclusion of specific HPV types that influence vaccine effectiveness adds depth to the understanding of these dynamics.

The study's insights, while subject to limitations in terms of study numbers and sub-group analyses, contribute substantively to the discourse surrounding HPV vaccination strategies. The findings underscore the potential benefits of simplifying vaccination regimens, particularly in resource-constrained settings. However, the decision to implement a one-dose regimen should be informed by extended follow-up studies and rigorous clinical outcome assessments. Ultimately, this investigation points to a path of enhanced vaccination coverage and improved accessibility, with the overarching goal of mitigating cervical cancer's global burden."

Reviewer #2: This is a very interesting topic, worthy of investigation.

However, I have a number of major comments:

- What is a one-dose HPV vaccine? There is no such vaccine. It would be better to speak of one-dose vaccination with an HPV vaccine.

- Note that there are 4 commercially available HPV vaccines, including the Chinese bivalent vaccine.

- Are the search terms sufficient? Should HPV and vaccine have been separated: HPV AND vaccine. Should “OR ‘single dose’ “ have been added, as an alternative for ‘one dose’? Generated an additional 130 hits on PubMed alone…

- Was the SLR/MA registered at PROSPERO? Nowadays, that should always be done.

- The authors claim to follow PRISMA guidelines but where is the PICOTS table?

- In the Risk of Bias paragraph in Results, the referencing is not consistent.

- The last sentence of paragraph 1 on page 13 is a duplication. [please use page numbers and line numbers next time]

- Study characteristics: 4 RCTs but 6 references.

- Page 14 second paragraph: vaccine effectiveness against what?

- Same paragraph, please explain why refs 38-40 are needed, they do not seem to be linked to HPV at all.

- Table 1. The PRISMA diagram says 11 studies were selected for MA, but only 7 are shown in table 1. In fact, the ones listed in Table 1 do not seem to overlap with the ones mentioned in the figures 4 – 10.

- Figures 4-10, please explain how the weight of the studies was derived, I couldn’t see the logic.

- Figure 5, can it be really true that the I2 is 0%? Yes, they all seem to be more or less in one line, but the 95Cis are enormous for some studies.

- Effectiveness against hrHPV – the vaccines were not designed to provide protection against NVT, does it make sense to discuss that here?

- Should ASC-H really be included with HSIL?

- Discussion – the authors do not really seem to discuss the results they obtained. I would have expected some remark about the observation that the effectiveness against clinical outcomes seems to be more similar between 1-dose vs 2(+)-dose than the effectiveness against infection…

- The authors state that more studies are needed but they fail to mention all currently ongoing one-dose trials: KEN-SHE, DORIS, ESCUDDO, and probably others. These studies are to be completed in the coming years, providing sufficient evidence to decide whether one-dose is effective.

- Overall, the manuscript suffers a language problem.

6. PLOS authors have the option to publish the peer review history of their article (what does this mean?). If published, this will include your full peer review and any attached files.

Reviewer #1: **Yes: **Sveinung Sørbye

Reviewer #2: **Yes: **Marc Baay

---

## [Author Response · Author response to Decision Letter 0]

30 Oct 2023

All of the comments, have been submitted by ms. word

---

## [Editor Report · Decision Letter 1]

8 Nov 2023

The Clinical Effectiveness of One-Dose HPV Vaccine: A meta-analysis of 902,368 vaccinated women

PONE-D-23-25664R1

Dear Dr. Didik Setiawan

We’re pleased to inform you that your manuscript has been judged scientifically suitable for publication and will be formally accepted for publication once it meets all outstanding technical requirements.

Kind regards,

Gregory Halle-Ekane, M.D;FWACS

Academic Editor

PLOS ONE

Additional Editor Comments (optional):

The comments and the suggestions of the reviewers have been adequately addressed as depicted in the rebuttal letter. We commend the authors.
---

## [Editor Report · Acceptance letter]

28 Dec 2023

PONE-D-23-25664R1 

PLOS ONE

Dear Dr. Setiawan, 

I'm pleased to inform you that your manuscript has been deemed suitable for publication in PLOS ONE. Congratulations! Your manuscript is now being handed over to our production team.

Kind regards, 

on behalf of

Professor Gregory Halle-Ekane 

Academic Editor

PLOS ONE